# Direct Current Stimulation Modulates Synaptic Facilitation via Distinct Presynaptic Calcium Channels

**DOI:** 10.3390/ijms242316866

**Published:** 2023-11-28

**Authors:** Sreerag Othayoth Vasu, Hanoch Kaphzan

**Affiliations:** Sagol Department of Neurobiology, University of Haifa, Haifa 3103301, Israel

**Keywords:** tDCS, calcium channels, axon terminals, subthreshold, weak electrical fields

## Abstract

Transcranial direct current stimulation (tDCS) is a subthreshold neurostimulation technique known for ameliorating neuropsychiatric conditions. The principal mechanism of tDCS is the differential polarization of subcellular neuronal compartments, particularly the axon terminals that are sensitive to external electrical fields. Yet, the underlying mechanism of tDCS is not fully clear. Here, we hypothesized that direct current stimulation (DCS)-induced modulation of presynaptic calcium channel conductance alters axon terminal dynamics with regard to synaptic vesicle release. To examine the involvement of calcium-channel subtypes in tDCS, we recorded spontaneous excitatory postsynaptic currents (sEPSCs) from cortical layer-V pyramidal neurons under DCS while selectively inhibiting distinct subtypes of voltage-dependent calcium channels. Blocking P/Q or N-type calcium channels occluded the effects of DCS on sEPSCs, demonstrating their critical role in the process of DCS-induced modulation of spontaneous vesicle release. However, inhibiting T-type calcium channels did not occlude DCS-induced modulation of sEPSCs, suggesting that despite being active in the subthreshold range, T-type calcium channels are not involved in the axonal effects of DCS. DCS modulates synaptic facilitation by regulating calcium channels in axon terminals, primarily via controlling P/Q and N-type calcium channels, while T-type calcium channels are not involved in this mechanism.

## 1. Introduction

Transcranial direct current stimulation (tDCS) is a subthreshold neurostimulation technique that is known to improve cortical functions [1,2,3]. In cortical areas, tDCS electrically affects the neurons that are located between strategically placed electrodes, altering their activity and excitability in a bidirectional manner [4]. Several clinical studies have suggested that this altered neuronal activity plays a crucial role in the modulation of cortical functions such as different aspects of cognition [1,5,6] and that bidirectional control of neuronal activity allows for both enhancement and reduction of cortical activity [2,3]. Numerous clinical studies demonstrate that tDCS ameliorates a wide range of medical conditions, including epilepsy, neurodegenerative diseases, neuropsychiatric disorders, and post-stroke conditions [4]. Thus, due to its wide range of applications, tDCS has recently gained popularity among researchers. However, despite various studies, the underlying mechanism of tDCS still remains unclear, limiting the efficacy and reliability of its therapeutic applications [1,7,8,9,10]. Especially, the relatively small external electric field is a major constraint in understanding the mechanism of tDCS, raising the question of how tDCS modulates cognitive functions despite the weak electric field intensity it generates.

The principal mechanism of tDCS-induced modulation is based on the polarization of cortical neurons. It is widely accepted that tDCS generates a small external electric current between the paired electrodes, anode, and cathode, causing the subcellular compartments of the neurons subjected between these electrodes to become differentially polarized [11,12,13,14]. Among the subcellular compartments, the sensitive axon terminals are strongly polarized because of their thin and elongated morphology. Previously, the cable equation was used to model the polarization of the axon terminals [12]. Recent results, however, show that this axon terminal polarization is much stronger than earlier predictions, and the increase in axon terminal polarization due to direct current stimulation (DCS) is attributed to the modification of ionic channels’ conductance [15,16]. Moreover, it was previously shown that by modifying axon terminal polarization, tDCS directly affects presynaptic vesicle release, a process that is critical for understanding the cognitive effects of tDCS [14,15,16]. Hence, further investigation of the mechanism by which tDCS affects presynaptic vesicle release in cortical neurons will enhance comprehension of how tDCS entails cognitive effects. 

Cognitive functions are a product of neural networks composed of individual neurons linked by synapses, where synapses are considered network nodes and are central to all cognitive functions, including learning, memory, and cognition [17,18,19,20]. To enable such cognitive functions, neurotransmitters are released into the synaptic cleft during neuronal activity. Nonetheless, even when the neurons are electrically inactive, axon terminals spontaneously release neurotransmitters into the synaptic cleft, which is essential for keeping the synaptic plasticity intact [21,22,23]. This spontaneous vesicle release process is not completely a stochastic event, but it is controlled by the baseline calcium concentration in the axonal compartment [24]. Noteworthy calcium channels are voltage-sensitive ionic channels that are crucial for membrane potential maintenance, action potential propagation, and information transfer [25,26]. There are a few types of calcium channels in neurons that are involved in regulating vesicle release [23,27], which probably play a role in neurostimulation. Their voltage-sensitive nature enables them to respond to subthreshold depolarization, and even a slight depolarization is sufficient to open the calcium channels and induce a presynaptic calcium influx that increases local calcium levels [28]. Presynaptic calcium is involved in controlling vesicle release from presynaptic boutons [24,29], through binding to proteins such as synaptotagmin, which facilitate the process of membrane fusion [30,31,32]. In an earlier study, we demonstrated that the P/Q-type calcium channels at the axon terminals projecting to the CA1 neurons are actively involved in the direct current stimulation (DCS)-induced modulation of spontaneous excitatory postsynaptic currents (sEPSCs) [33]. This coincides with previous studies that reported the abundance of P/Q-type calcium channels at the axon terminals [34,35]. However, our earlier study investigated hippocampal CA1 neurons and not cortical neurons that are more relevant for tDCS and did not consider other calcium channels that might play a role in tDCS effects [33]. Additionally, inputs to the CA1 neurons, the EC cortex outputs, and the Schaffer collateral circuitry are both very long and form bundles, giving them directional specificity. This elongated nature causes the axon terminals to become highly polarized, making the hippocampal neurons an optimal target for tDCS. However, in the motor cortex, layer-V inputs are mainly from layer-II/III neurons, and contrary to CA1, layer-V neurons are spread throughout a large region of the cortex rather than being grouped together [36]. In order to reach their widely dispersed layer-V neurons, the branchlets of layer-II/III neurons deviate from the parent axon, making their axons more varied and arborized in character. The DCS-induced terminal polarization may be reduced by layer-II/III arborization. Given that the P/Q and N calcium channels have high thresholds, we wondered if the external polarization was strong enough to produce a DCS-induced effect. Nonetheless, we posited that calcium channels play a role in tDCS in the cortex since a previous study showed that the calcium channel antagonist fluranizine reduced tDCS-induced cortical excitability in the human motor cortex [8], but the underlying mechanism of this effect was not clear. As a result, we decided to expand the experiment to include more distinct calcium channels in the motor cortex in order to gain a more comprehensive view of the role of calcium channels in the cortical region. Therefore, in order to examine the active participation of calcium channels in the DCS-induced vesicle release modulation in cortical neurons, we recorded sEPSCs from layer-V pyramidal neurons in the motor cortex while selectively inhibiting distinct calcium channels. Thus, we demonstrated that calcium channels are important for the modulation of synaptic vesicle release by DCS and delineated the specific subtypes of calcium channels that play a crucial role in this process. In particular, we highlight the relevance of P/Q and N-type calcium channels in this regard.

## 2. Results

### 2.1. DCS-Induced Modulation of sEPSCs Is Not Mediated by T-Type Calcium Channels 

In our previous study, we demonstrated that the DCS application to cortical slices in the M1 motor cortex strongly modulates sEPSC frequency, so that anodal stimulation that depolarizes the axons enhances the presynaptic vesicle release rate, while cathodal stimulation that hyperpolarizes the axons produces the opposite effect of reducing the rate of spontaneous vesicle release (Appendix A) [15]. In another study, we showed that this is also true in the hippocampus, where DCS modulates sEPSC frequency by altering P/Q-type calcium channel conductance. There, we demonstrated that in hippocampal slices, DCS effects were completely occluded by inhibiting P/Q-type calcium channels [33]. It is well established that baseline calcium concentrations modulate sEPSCs, and even minor variations in the axonal calcium concentrations affect spontaneous vesicle release [32,37]. In neurons, calcium concentration is regulated by various types of calcium channels. Among these, T-type calcium channels are the major low-threshold calcium channels, known for their activation during subthreshold depolarization [38,39,40]. Moreover, the cortex is the major brain region affected by tDCS, and T-type calcium channels are abundant in cortical neurons. Therefore, due to their low-threshold activation and high expression in the cortex, we posited that the T-type calcium channels are significant players in modulating presynaptic vesicle release by tDCS in cortical circuits, and we expected that inhibiting these channels would occlude DCS effects on vesicle release. To test this, we recorded sEPSCs from layer-V pyramidal neurons at the motor cortex and examined the effect of T-type calcium channel inhibition using ML-218 500 nM perfused via the bath (Figure 1A,B). Surprisingly, T-type calcium channel inhibition did not occlude DCS effects. Despite ML218 application, anodal DCS enhanced sEPSC frequency while cathodal DCS reduced the sEPSC frequency [F_(1.30, 37.80)_ = 22.05, *p* < 0.01 in RM-ANOVA; t_(29)_ = 5.78, *p* < 0.0001 and t_(29)_ = 2.9, *p* = 0.02 in posthoc Bonferroni corrected comparison between anodal and no-DCS and cathodal and no-DCS, respectively] (Figure 1C,D). Likewise, under the application of ML-218, the cumulative distribution of the interevent intervals was observed to shift leftwards during anodal condition and the cumulative distribution of the interevent intervals was observed to shift rightwards under cathodal condition [D = 0.09, *p* < 0.0001 and D = 0.06, *p* = 0.005 in K-S between anodal DCS to no-DCS and between cathodal and no-DCS, respectively] (Figure 1E). Intriguingly, under the application of ML-218, sEPSCs amplitude was affected only by anodal DCS but not by cathodal DCS, meaning that modulation of sEPSCs amplitude by cathodal DCS was occluded by T-type calcium channel inhibition, but anodal DCS was not affected by the same inhibition. During the bath perfusion of ML-218, the anodal DCS induced a significant enhancement in the sEPSCs mean amplitudes; however, cathodal DCS did not induce any significant variation in the mean sEPSCs amplitudes [F_(1.88, 54.76)_ = 5.96, *p* = 0.005 in RM-ANOVA; t_(29)_ = 3.10, *p* = 0.01 and t_(29)_ = 0.60, *p* > 0.99 in posthoc Bonferroni corrected comparison between anodal and no-DCS and cathodal to no-DCS, respectively] (Figure 1C,F). In addition, the cumulative distribution of the sEPSCs amplitudes was observed to shift rightwards during the anodal DCS application, while the cumulative distribution of the sEPSCs amplitudes during cathodal DCS was homogenous to the no-polarization condition [D = 0.14, *p* < 0.0001 and D = 0.03, *p* = 0.19 in K-S between anodal-DCS and no-DCS and between cathodal and no-DCS, respectively] (Figure 1G). These results were in contrast to our initial anticipation and demonstrated that T-type calcium channels are not involved in the modulation of DCS-induced presynaptic vesicle release. Nonetheless, this is not completely surprising, considering that there are no reports of T-type calcium channels at the distal end of the axons, where tDCS effects are the strongest [12,13,14], and they were not shown to be actively involved in the process of spontaneous vesicle release [39]. Most of the T-type calcium channels were observed to occupy the dendrites and the soma [41]. Taken together, we conclude that T-type calcium channels are not responsible for the reported modulation of presynaptic vesicle release frequency, even though they are active in the subthreshold range. Moreover, the results herein revalidate the importance of the polarization of axon terminals during DCS-induced synaptic vesicle release.

### 2.2. P/Q-Type Calcium Channels Play a Role in DCS-Induced Modulation sEPSCs

As aforementioned, we showed that T-type calcium channels are not actively involved in the DCS-induced sEPSC frequency modulation. It is well known that P/Q-type calcium channels are the main subtype of calcium channels that are expressed in neurons and are involved in synaptic vesicle release [34,35,42]. In our previous study, we demonstrated that in the hippocampus, P/Q-type calcium channels are crucial for DCS-induced spontaneous release [33]. However, this study was limited to the recordings of CA1 neurons in the hippocampus receiving inputs from the perforant pathway and the Schaffer collaterals. Therefore, we extended our study to validate the role of P/Q-type calcium channels in DCS-induced modulation of spontaneous vesicle release in the motor cortex. Towards that, we recorded sEPSCs from motor cortex layer-V pyramidal neurons under different DCS conditions while blocking voltage-dependent P/Q-type calcium channels using ω-agatoxin 400 nM applied in the bath (Figure 2A,B). As predicted, ω-agatoxin completely occluded DCS effects. Both anodal and cathodal-DCS, under the application of ω-agatoxin, did not induce any significant alterations in the mean sEPSC frequency in comparison to no-DCS sEPSC frequency [F_(1.701, 35.73)_ = 9.89, *p* < 0.01 in RM-ANOVA; t_(21)_ = 2.46, *p* = 0.07, and t_(21)_ = 2.32, *p* = 0.09 in posthoc Bonferroni corrected comparison between anodal to no-DCS and cathodal to no-DCS, respectively] (Figure 2C,D). Nonetheless, the highly sensitive K-S test of cumulative distributions of the sEPSC interevent intervals still demonstrated a leftward shift with anodal DCS while cathodal DCS shifted rightwards [D = 0.10, *p* = 0.002 and D = 0.10, *p* = 0.0002 in K-S between anodal-DCS and no-DCS and between cathodal and no-DCS, respectively] (Figure 2E). Similarly, DCS application, under ω-agatoxin administration, did not alter the mean sEPSC amplitudes. The mean sEPSCs amplitudes under anodal and cathodal-DCS were comparable to those under no-DCS condition [F_(1.284, 26.97)_ = 7.927, *p* = 0.006 in RM-ANOVA; t_(22)_ = 2.40, *p* = 0.08, and t_(22)_ = 2.26, *p* = 0.1 in posthoc Bonferroni corrected comparisons between anodal and no-DCS and between cathodal and no-DCS, respectively] (Figure 2F). However, again, the more sensitive statistical analysis of the cumulative distributions of sEPSCs amplitudes under anodal and cathodal DCS application during ω-agatoxin administration was different than in the no-DCS condition [D = 0.18, *p* < 0.0001 and D = 0.04, *p* = 0.0005 in K-S for both anodal and no-DCS and between cathodal and no-DCS], and ω-agatoxin administration was not able to completely occlude DCS effects (Figure 2G). These results differ from our previous study in the hippocampus, where the more sensitive tests of cumulative distribution analyses showed a complete occlusion of DCS effects under ω-agatoxin administration [33]. Yet, although weaker compared to the CA1 neurons, P/Q-type calcium channels play a role in DCS-induced neuromodulation of synaptic functioning in the motor cortex, at least to some extent. Therefore, despite the regional heterogeneity of the CNS, our results demonstrate the functional significance of P/Q-type calcium channels in DCS-induced sEPSC modulation.

### 2.3. N-Type Calcium Channels Are Crucial for DCS-Induced Modulation of sEPSCs 

As abovementioned, we demonstrated that the P/Q-type calcium channels are involved in DCS-induced modulation of spontaneous vesicle release. However, along with the P/Q type calcium channels, N-type calcium channels are also known for their involvement in the synaptic vesicle release process from axon terminals [42,43,44]. Despite their moderate response to subthreshold depolarization, N-type calcium channels are known to be involved in subthreshold neuromodulation [43,44]. Therefore, to examine the role of N-type calcium channels in DCS-induced modulation of spontaneous vesicle release, we recorded sEPSCs from layer-V pyramidal neurons in the motor cortex under DCS stimulation while blocking N-type calcium channels using ω-conotoxin 500 nM perfused in the bath (Figure 3A,B). Under the application of ω-conotoxin, both anodal and cathodal-DCS did not induce any significant alterations in the mean sEPSCs frequency in comparison to no-DCS, and it was evident that ω-conotoxin application abolished any DCS-induced modulation of spontaneous vesicle release [F_(1.40, 28.06)_ = 1.30, *p* = 0.18 in RM-ANOVA; t_(20)_ = 1.35, *p* = 0.58; and t_(20)_ = 0.85, *p* > 0.99 in posthoc Bonferroni corrected comparisons between anodal to no-DCS and between cathodal to no-DCS, respectively] (Figure 3C,D). However, the highly sensitive test of comparing the cumulative distributions of sEPSC interevent intervals revealed an asymmetric effect of ω-conotoxin administration during DCS application. Anodal DCS induced a leftward shift of the cumulative distribution of sEPSC interevent intervals despite ω-conotoxin administration, while the rightward shift of the cumulative distribution of the interevent intervals that is usually induced by cathodal DCS was completely occluded by ω-conotoxin and became comparable to that of the no-DCS condition [D = 0.09, *p* = 0.001 and D = 0.06, *p* = 0.08 in K-S for both between anodal and no-DCS and between cathodal and no-DCS] (Figure 3E). Interestingly, ω-conotoxin administration during DCS application had a strong differential effect on sEPSC amplitudes. While ω-conotoxin administration did not occlude the enhancement of sEPSCs amplitudes by anodal DCS, it completely abolished cathodal-DCS effects, and cathodal DCS under ω-conotoxin administration did not have any effect on sEPSCs amplitude compared to no-DCS condition [F_(1.09, 21.80)_ = 11.37, *p* = 0.002 in RM-ANOVA; t_(20)_ = 3.21, *p* = 0.01 and t_(20)_ = 0.85, *p* = 0.55 in posthoc Bonferroni corrected comparisons between anodal to no-DCS and between cathodal to no-DCS, respectively] (Figure 3F). This effect was also observed with the cumulative distribution of the sEPSC amplitudes (Figure 3G). Despite ω-conotoxin application, anodal DCS induced a strong rightward shift in the cumulative distribution of the sEPSCs amplitudes, while the cumulative distribution of the sEPSCs amplitudes under cathodal DCS with ω-conotoxin application was comparable to the no-DCS condition [D = 0.22, *p* < 0.0001 and D = 0.02, *p* = 0.61 in K-S for both between anodal and no-DCS and between cathodal and no-DCS] (Figure 3G). These results demonstrate that along with the P/Q type calcium channels, N-type calcium channels also play a crucial role in the DCS-induced modulation of synaptic vesicle release while also playing a unique and asymmetric postsynaptic role.

## 3. Discussion

Transcranial direct current stimulation is a subthreshold stimulation that modifies cognitive functioning [1,2,3]. This is accomplished in part by modulating the neuronal activity of the cortical region. Previous research has shown that tDCS modifies neuronal function by altering the excitability and activity of cortical neurons [2,3]. Furthermore, tDCS is gaining a lot of attention for its ability to ameliorate a large spectrum of neuropsychiatric disorders such as neurodegenerative diseases, depression, post-stroke conditions, and many others [4]. Despite numerous studies, the underlying cellular mechanism is not completely clear, raising a major question on how the weak DC field induces the reported clinical changes [1,7,8,9,10]. This lack of understanding of its mechanism is the major hindrance to the acceptance of tDCS as a reliable therapeutic application by the clinical community. 

The principal mechanism behind tDCS-induced neuromodulation is that the external subthreshold electric field differentially polarizes the neuronal compartments, maximally when the axo-dendritic axis is in parallel to the direction of the electrical field. Among the neuronal compartments, axon terminals are more sensitive to the external electric field due to their elongated, thin nature [12,13,14]. Nonetheless, it initially seemed that the passive polarization induced by the external electric field was not substantial enough to produce modifications in the neuronal network and cause cognitive and clinical changes [12,13]. However, recent studies illustrate that the axon terminal polarization is stronger than earlier predictions [14,15]. This is due to the active involvement of ionic channels such as the sodium and potassium channels that amplify the terminal polarization, which in turn affects the synaptic vesicle release from the presynaptic compartment [15,33]. 

Cognition is based upon neuronal network functioning, which is made up of multiple interconnected individual neurons [17,18,45]. The presynaptic compartments of these neurons release neurotransmitters that connect between neurons [22,27,46,47]. Therefore, in order to understand the influence of DCS on network functioning, we examined its effects on presynaptic release modulation. In our previous studies, we already showed that anodal DCS that depolarizes axon terminals increases sEPSC frequency, while cathodal DCS that hyperpolarizes the axon terminals decreases sEPSC frequency both in the motor cortex (Appendix A) and in the hippocampus, although these effects are not completely symmetric and the depolarizing effects are a bit stronger [15,33]. Our experiments herein show that DCS modulates sEPSC frequency by controlling the activity of calcium channels. sEPSC represents the event of synaptic vesicle release from the presynaptic compartment. Therefore, this DCS-induced modulation of sEPSC frequency signifies the ability of DCS to modulate presynaptic vesicle release. Noteworthy synapses have constant background activity even when no action potential is conducted along the axon, and spontaneous events of vesicle release from presynaptic terminals contribute to keeping synaptic connections intact [37,48]. This mechanism is essential for cognitive functions [49]. Activity-driven synaptic vesicle release is also regulated by the local calcium concentration in the axon terminal compartments [25,27]. Likewise, spontaneous vesicle release is not a mere stochastic function; it is tightly linked to the baseline calcium concentrations at the presynaptic boutons, and alterations in calcium concentration modify the presynaptic vesicle release [50]. As indicated in our prior research, polarization of the axon terminals of long axons bidirectionally modifies the rate of spontaneous vesicle release from the presynaptic compartments, so depolarization by anodal DCS increases sEPSC frequency, while hyperpolarization by cathodal DCS reduces sEPSC frequency (Appendix A) [15,16,33]. We also showed that cutting the axons and making them much shorter abolished the spontaneous vesicle release modulation by DCS [33]. Taken together, these data show that axon terminals are very sensitive to DCS and that, if long enough, DCS induces a physiologically significant membrane polarization at the axon terminals. This polarization of the axon terminals opens calcium channels and increases baseline calcium levels, which in turn modifies the vesicle release probability from the presynaptic compartments. Even a small subthreshold variation in the membrane potential is enough to alter the intracellular calcium concentration [28]. Here, we demonstrate that by inhibiting P/Q or N-type calcium channels, we occlude the DCS-induced modulation of presynaptic vesicle release, suggesting that DCS-induced polarization alters the calcium channel conductivity and in turn modifies the vesicle release from the presynaptic compartment. P/Q and N-type calcium channels are the most common subtypes of calcium channels found in the axon terminals of cortical neurons [34,35,42,43]. By co-localizing with synaptotagmin-rich vesicle clusters, both P/Q and N-type channels play an important role in spontaneous vesicle release [32]. They control the local calcium concentrations and facilitate presynaptic vesicle release by helping in the vesicle fusion process [32]. Noteworthy, though sporadic in the axon terminals, other calcium subtypes, such as R-type channels, are known to not participate in spontaneous vesicle release [42]. However, N-type and P/Q-type calcium channels are the major channels present in the axon terminal that are vital for synaptic vesicle release. L-type channels are not present in the axon terminals in the motor cortex. Furthermore, it is well established that both P/Q and N-type channels aid in the synaptic facilitation process, in addition to their contribution to membrane depolarization, and assist synaptic vesicles in fusing with the membrane [51]. However, L-type channels do not play a crucial role in synaptic vesicle release. Furthermore, the abovementioned results also demonstrate the asymmetry of DCS-induced synaptic modulation. This is in line with the notion that anodal and cathodal DCS applications have different degrees of effectiveness. Previous research has shown that anodal DCS application has stronger effects than cathodal DCS application [5,15,52,53]. This slight asymmetry is also evident in our experiments, particularly when ω-conotoxin is applied via bath. Despite the ω-conotoxin administration, the cumulative distribution of interevent intervals showed a slight leftward shift with anodal DCS application (Figure 3E). Yet, this phenomenon was not observed during cathodal DCS application, wherein the cumulative distribution of the interevent intervals with cathodal DCS and ω-conotoxin was comparable to the no DCS condition, showing a complete occlusion of cathodal DCS effects with ω-conotoxin (Figure 3E). However, note that this minor variation in the interevent intervals is insufficient to cause a significant change in the mean sEPSC frequency. The same point is also demonstrated by a similar asymmetry in the sEPSC amplitudes. In addition, our results suggest the possibility that the two types of P/Q and N-type calcium channels work in concert, as inhibiting one type of channel occludes almost all of the presynaptic effects of DCS, but the very sensitive statistical comparisons of the cumulative distributions of sEPSCs interevent intervals (Figure 2E and Figure 3E) show that DCS effects are not completely abolished, probably due to the residual effects of the non-inhibited calcium channels. As mentioned above, DCS-induced polarization is highest at the axon terminals. Although not tested in the herein study, given that both R- and L-type calcium channels are very sparse in the axon terminals in the motor cortex, their contribution to the axon terminal polarization is probably minimal, if any. However, it is well established that activation of P/Q and N-type channels that are abundant in the axon terminals results in the influx of calcium ions, further depolarizing the axon terminals. Moreover, P/Q (Cav2.1) and N-type (Cav2.2) calcium channels show strong co-localization with synaptotagmin-containing vesicle clusters, while the R-type channel (Cav2.3) does not involve or only partially co-localize with synaptotagmin-containing vesicle clusters and was not shown to play a significant role in synaptic vesicle fusion [23,54,55]. Previous studies demonstrated that the P/Q type and N type channels are vital for synaptic vesicle release modulation, and despite being high voltage-gated channels, both P/Q-type and N-type calcium channels link even relatively small neuronal depolarization to neurotransmitter release [44]. In summary, we show that DCS polarizes the axon terminal membrane potential, enough to induce substantial fluctuations in the resting baseline calcium levels at axon terminals via controlling either P/Q or N-type calcium channel conductivity, suggesting that each of these channels is crucial for the process of DCS-induced modulation of the spontaneous vesicle release. 

Unlike P/Q and N-type calcium channels, which are high threshold-activated channels, T-type calcium channels are low voltage-activated calcium channels and are well known for their interaction in subthreshold modulation [38,40]. They are mostly found in the dendrites and soma and aid in signal propagation, membrane potential maintenance, and burst firing [38,40,56]. Due to their ability to activate at low thresholds, T-type calcium channels are highly susceptible to responding to tDCS-induced subthreshold depolarization and being involved in its underlying mechanism. However, the T-type channels are transient and inactivate rapidly upon depolarization, which makes them less effective for mediating the effects of the long stimulation of DCS [41]. Moreover, most T-type calcium channels are seen to be found in the dendrites and the soma [39] and seldom localize to the distal end of axons in presynaptic terminals, although this was reported in some cases and they were shown to contribute to vesicle release [57]. Overall, these coincide with our findings that despite their activity in the subthreshold range, T-type calcium channels did not occlude the observed DCS-induced modulation of sEPSC frequency (Figure 1E), meaning that their inhibition does not prevent the presynaptic effects of DCS. Nonetheless, we observed a postsynaptic effect of T-type calcium channel inhibition on cathodal DCS, as it prevented the cathodal DCS-induced reduction in sEPSC amplitude (Figure 1F,G). The effect of the T-type calcium channel blocker ML218 is partially irreversible [58], and the reversibility of the drug effect was not investigated in the present study.

A possible caveat to our study is the issue of space clamps in our voltage-clamp recordings, which might affect to some extent the recording of sEPSC amplitudes that are from distant dendrites. Nonetheless, potential variations in distant dendrites are not well recorded in voltage-clamps anyhow, and our main interpretation of the results concerned the sEPSC frequency and not the amplitudes, which were merely reported. Another possible limitation to this study is that other neuromodulation mechanisms, such as DCS-induced effects on astrocytes, may also be important in the DCS-induced modulation of sEPSCs. However, since astrocytes are quite symmetric and not polarized cells, it is expected that their sensitivity to polarization would be much less than that of pyramidal neurons. Nonetheless, since they have prolonged thin processes, it is expected that the membrane of the processes that are oriented along the electrical field direction will be polarized, but this effect would be symmetrical, half depolarized, and half hyperpolarized. Noteworthy, previous studies have shown that prolonged application of tDCS enhanced the calcium activity in astrocytes, and this could possibly modulate neuronal activity and synaptic plasticity [59,60].

Taken together, our findings show that DCS-induced calcium regulation contributes to synaptic vesicle release in the motor cortex. Please take note that previous studies have shown that the DCS treatment polarizes the axon terminals [12,61]. This axon terminal polarization allows calcium ions to enter the compartment and further depolarize the terminal by opening the P/Q and N-type calcium channels that are present there. An earlier study found that the calcium channel antagonist fluranizine reduced tDCS-induced cortical excitability in the human motor cortex [8]. Although the precise mechanism of tDCS-induced regulation of cortical activity is unclear, we suggest that variations in baseline calcium levels caused by DCS affect synaptic vesicle release, which in turn affects the excitability of cortical neurons. Additionally, the quantity of calcium present in neurons is a key determinant of synaptic plasticity and may be crucial for the long-term effects of tDCS [62]. Therefore, our research suggests that the potential interaction of tDCS with calcium channel blockers, which are sometimes prescribed as medications for certain conditions, might interfere with the beneficial effects of tDCS when co-applied as a therapy. 

To conclude, DCS modulates synaptic facilitation through regulating calcium channels in axon terminals, mainly via controlling P/Q and N-type calcium channels, which play a significant role in this process. In addition, due to their unique characteristics, T-type calcium channels are not involved in the modulation of presynaptic vesicle release by DCS. We also show that the ongoing synaptic plasticity during the resting state and the maintenance of connectivity can also be affected by DCS, as it increases spontaneous synaptic activity. Modulating ongoing synaptic plasticity is essential for modifying cognitive functions. Taken together, it is likely that the regulation of cognitive performance by tDCS is dependent on its ability to modulate calcium channel conductance. 

## 4. Materials and Methods

### 4.1. Animals

The subjects of the experiments were older than 2 months old and were all male C57/Bl6 mice. By this age, most of the developmental activity in a mouse brain is considered complete, and there are no significant changes in the structure or morphological properties of neurons in its CNS. The effectiveness of DCS treatment is primarily determined by axonal properties such as axon length and myelination. The axon length and myelin thickness differ significantly between genders [63,64]. Furthermore, previous research suggests that the current delivered to neurons during tDCS also differs by gender [65,66]. To avoid possible sex-dependent confounds, only male mice were used. Animals were bred in the animal facility with a 12 h light-dark cycle. Water and food were available ad-libitum. Housing, handling, and experimental procedures were performed in accordance with the National Institutes of Health guidelines and were approved by the University of Haifa animal ethics committee. 

### 4.2. Electrophysiology

The whole-cell rig was based on an Olympus microscope BX51-WI (Tokyo, Japan) with a 60× objective (NA = 1.0), and an oil condenser (NA = 1.4). The camera for DIC imaging and fluorescent imaging is the DAGE-MTI IR-1000E (DAGE-MTI, Michigan City, IN, USA). Electrophysiological recordings were acquired using a Multiclamp 700B amplifier with a Digidata 1440 digitizer, run by pclamp10.7 software (all by Molecular devices, Sunnyvale, CA, USA).

### 4.3. Slices Production for Electrophysiology

Slices were produced by cutting with a Campden vibratome 7000smz-2 that has Zero Z technology to minimize *z*-axis deflection. Animals were cervically dislocated, and their brains were taken immediately to harvest coronal slices (300 µm) using the Campden Vibratome 7000 smz2 (Campden Instruments, Leicestershire, UK) in the fluff of frozen slicing solution (in mM): 110 Sucrose, 60 NaCl, 3 KCl, 1.25 NaH_2_PO_4_, 28 NaHCO_3_, 0.5 CaCl_2_, 7 MgCl_2_, and 5 Glucose. Slices were kept for incubation at 34.5 °C for ~1.5 h in artificial cerebrospinal fluid (ACSF) containing in mM: 125 NaCl, 2.5 KCl, 1.25 NaH_2_PO_4_, 25 NaHCO_3_, 25 D-glucose, 2 CaCl_2_, and 1 MgCl_2_. An additional recovery period of 30 min at room temperature was given inside the electrophysiology chamber in ACSF (2 mL/min).

### 4.4. Voltage Clamp Whole-Cell Recording

Slices were kept for 30 min in an electrophysiology rig for recovery while caboxygenated with 95% O_2_ + 5% CO_2_. Brain slices were observed using an objective ×60 with an NA = 1 mounted on a whole-cell rig microscope (BX51-WI, Olympus, Center Valley, PA, USA), and the images were captured with a charge-coupled device camera IR-1000 (Dage MTI, Michigan City, IN, USA). Signals from the electrode were amplified using Multiclamp 700B and digitized using Digidata 1440 (molecular devices, Sunnyvale, CA, USA). Borosilicate glass pipettes (3–5 MΩ for soma) were pulled using a pipette puller (P-1000; Sutter Instruments, Navato, CA, USA) and filled with a K-gluconate-based internal solution (in mM): 120 K-gluconate, 20 KCl, 10 HEPES, 2 MgCl_2_, 4 Na_2_ATP, 0.5 TrisGTP, 14 phosphocreatine, osmolarity 290 mOsm, and pH = 7.3. Whole-cell recordings were performed according to standard procedure. For the synaptic current measurements, sEPSCs were recorded in a voltage clamp from the somas of pyramidal cells in the region of the Motor cortex at a holding potential of −70 mV using the K-Gluconate-based internal solution, as previously described [67,68]. Here, internal solutions based on K-gluconate are employed to mimic physiological conditions, whereas Cs-based internal solutions are avoided to prevent depolarization at distal regions brought on by potassium blockade, which may interact with DCS-induced polarization at these terminal regions. We recorded from pyramidal cells that their dendro-axonic axis was perpendicular to the orientation of the wires, meaning it was parallel to the applied DC field (Figure 1B). All voltage-clamp recordings were low-pass filtered at 10 kHz and sampled at 50 kHz. Series resistance was 90% compensated. Series resistance, input resistance, and membrane capacitance were monitored during the entire experiment. Data exclusion criteria were based on changes in the above parameters along the experiment of more than 15% from the baseline. The experiments were performed at a constant 21 °C room temperature. We started the recording of the sEPSCs a minute after the application of DCS. During this time, any fluctuations are stabilized. For analysis, we took the middle 30 s of the recording. To avoid any possible fast noise perturbations in the results, a lower cutoff of 8 pA was set, and sEPSCs with amplitudes lower than 8 pA were excluded from analysis. To avoid action potential being included in the measurements of sEPSCs, we introduced an upper cutoff of 100 pA. For each recording of every neuron, we selected an appropriate sEPSC template according to the clampfit software protocol; thus, possible additional artifacts were also excluded. Furthermore, sEPSC encompasses stochastic events like mEPSCs as well as depolarization-dependent events, and we aimed to include depolarization-dependent events in our study since the principal effects of DCS rely on depolarization and hyperpolarization. In our previous study, we demonstrated that sodium channels are essential for DCS-induced active polarization of axon terminals and subsequent spontaneous vesicle release modulation. Furthermore, blocking sodium channels with TTX resulted in the abolition of DCS-induced sEPSC modulation [15]. Noteworthy, sodium channel inhibition also blocked the sodium persistent current, which in itself slightly hyperpolarizes the axon terminals. In addition, it prevents the active component of DCS-induced depolarization, which is dependent on the sodium channel [15]. Therefore, to study DCS-induced synaptic vesicle release modulation, we cannot use TTX. 

### 4.5. DC Field

The uniform DC electric field is generated by a linear stimulus isolation unit (LSIU-02, Cygnus Technology, Delaware Water Gap, PA, USA), and the cathode and anode were made using AgCl-covered silver wires (783500, AM-System, Carlsborg, WA, USA). The recording chamber held a contraption that produced a homogenous linear electrical field. The contraption contained two thick (2 mm) Ag-AgCl wires that were parallel to each other, and the brain coronal slices were placed in the middle between the two wires. The Ag-AgCl wires were connected to the two poles of the LSIU to produce a homogenous linear electrical field. The parallel orientation of the Ag-AgCl wires and their thickness (much more than the slices) enabled the external electric field not to vary in the axis orthogonal to the applied field (along the wires) or in the *z*-axis of the slice (vertically within the 300 μm thickness of the slice in a septo-temporal axis). The applied DCS voltage is examined before each experiment to ascertain a uniform, homogenous, and linear electric field (EF) of exactly 5 V/m in the *x*-axis (between the wires) and exclude variations in the *y* (along the wires) or *z* axes. The depolarizing, hyperpolarizing, and no-polarizing electrical field conditions were applied from the beginning of each experiment to its end.

### 4.6. Statistics

GraphPad Prism 7.05 (GraphPad Software, San Diego, CA, USA). A one-way and repeated measure ANOVA is used for electrophysiological data analysis, with *p* < 0.05 as the significance criteria for the Bonferroni post hoc test. Cumulative distributions of sEPSCs were performed without binning. F values and their corresponding degrees of freedom are reported in the repeated measures ANOVA for the three groups (anodal, n-DCS, and cathodal), followed by posthoc *t*-tests between each of the two groups, with Bonferroni correction for multiple comparisons, and the t-values and their corresponding degrees of freedom are also reported. In addition, Kolmogorov–Smirnov (K-S) tests were used to compare the differences between each of the two cumulative distributions, and D values for the K-S tests are reported. Outliers of sEPSCs with interevent intervals >1500 and amplitudes >100 pA were excluded.

## Figures and Tables

**Figure 1 ijms-24-16866-f001:**
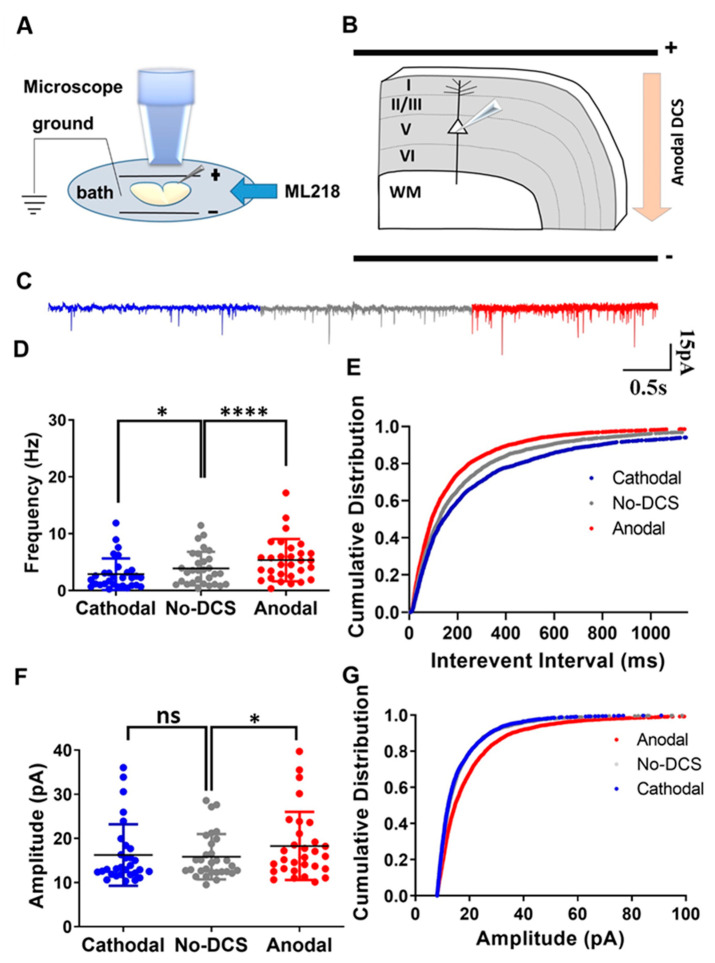
T-type calcium channels are not involved in the DCS-induced modulation of presynaptic vesicle release. (**A**) Illustration of the experimental setup. Pyramidal neurons from layer-V M1 motor cortex were recorded when DCS (5 V/m) was applied across the slice and ML218 was perfused via bath. (**B**) The diagram depicts the direction of the electrical field with respect to the orientation of the recorded layer-V cortical pyramidal neurons, via an example of anodal DCS application. (**C**) Sample traces for sEPSCs recorded under anodal, cathodal, and no DCS conditions are displayed in red, blue, and grey colors, respectively. (**D**) Under ML218 application, anodal DCS enhanced mean sEPSCs frequency while cathodal DCS reduced the mean sEPSCs frequency [F_(1.30, 37.80)_ = 22.05, *p* < 0.01 in RM-ANOVA; t_(29)_ = 5.78, *p* < 0.0001 and t_(29)_ = 2.9, *p* = 0.02 in post hoc Bonferroni-corrected comparison between anodal and no-DCS and cathodal to no-DCS, respectively] Data are presented as Mean ± SD. (**E**) The cumulative distribution of the interevent intervals was observed to shift leftwards during anodal condition and rightwards under cathodal condition under the application of ML-218 [D = 0.09, *p* < 0.0001 and D = 0.06, *p* = 0.005 in K-S between anodal-DCS and no-DCS and between cathodal and no-DCS, respectively]. (**F**) Under the bath perfusion of ML-218, the anodal DCS induced a significant enhancement in the sEPSCs mean amplitudes; however, cathodal DCS did not induce any significant variation in the mean sEPSCs amplitudes [F_(1.88, 54.76)_ = 5.96, *p* = 0.005 in RM-ANOVA; t_(29)_ = 3.10, *p* = 0.01 and t_(29)_ = 0.60, *p* > 0.99 in posthoc Bonferroni corrected comparison between anodal to no-DCS and cathodal to no-DCS, respectively]. Data are presented as Mean ± SD. (**G**) The cumulative distribution of the sEPSCs amplitudes was observed to shift right-wards during the anodal DCS application, while the cumulative distribution of the sEPSCs amplitudes during cathodal DCS was homogenous to the no-polarization condition [D = 0.14, *p* < 0.0001 and D = 0.03, *p* = 0.19 in K-S between anodal-DCS and no-DCS and between cathodal and no-DCS, respectively]. N = 6 mice, n = 30 neurons for all panels. A maximum of two neurons were recorded per slice. * *p* < 0.05; **** *p* < 0.0001; ns = non-significant.

**Figure 2 ijms-24-16866-f002:**
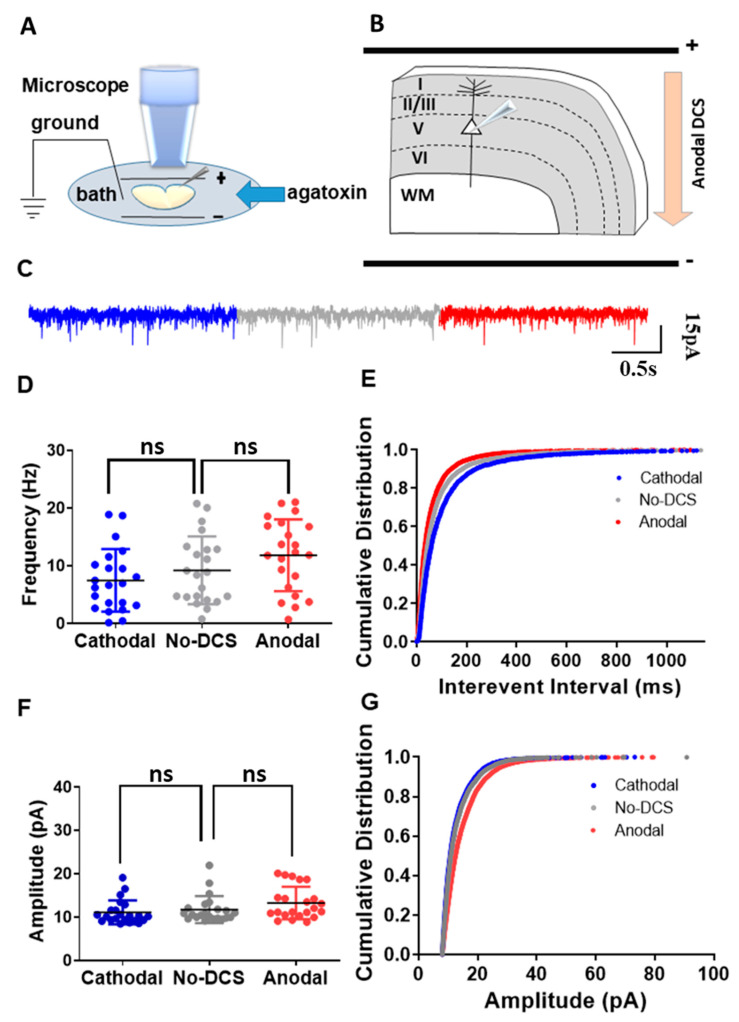
P/Q-type calcium channels are involved in the DCS-induced modulation of presynaptic vesicle release. (**A**) Illustration of the experimental setup. Pyramidal neurons from layer-V M1 motor cortex were recorded when DCS (5 V/m) was applied across the slice and ω-agatoxin was perfused via bath. (**B**) The diagram depicts the direction of the electrical field with respect to the orientation of the recorded layer-V cortical pyramidal neurons, via an example of anodal DCS application. (**C**) Sample traces for sEPSCs recorded under anodal, cathodal, and no DCS conditions are displayed in red, blue, and grey colors, respectively. (**D**) The application of ω-agatoxin did not induce any significant alterations in the mean sEPSC frequency during anodal and cathodal DCS conditions in comparison to no-DCS sEPSC frequency [F_(1.701, 35.73)_ = 9.89, *p* < 0.01 in RM-ANOVA; t_(21)_ = 2.46, *p* = 0.07, and t_(21)_ = 2.32, *p* = 0.09 in posthoc Bonferroni corrected comparison between anodal and no-DCS and cathodal to no-DCS, respectively] Data are presented as Mean ± SD. (**E**) The more sensitive test of K-S demonstrated that cumulative distributions of the sEP-SCs interevent intervals of anodal DCS shift leftwards while cathodal DCS shift rightwards [D = 0.10, *p* = 0.002 and D = 0.10, *p* = 0.0002 in K-S between anodal-DCS to no-DCS and between cathodal and no-DCS, respectively]. (**F**) The mean sEPSCs amplitudes under anodal and cathodal-DCS were comparable to those under no-DCS condition under the application of ω-agatoxin [F_(1.284, 26.97)_ = 7.927, *p* = 0.006 in RM-ANOVA; t_(22)_ = 2.40, *p* = 0.08, and t_(22)_ = 2.26, *p* = 0.1 in posthoc Bonferroni corrected comparisons between anodal and no-DCS and between cathodal and no-DCS, respectively]. Data are presented as Mean ± SD. (**G**) The cumulative distributions of sEPSC amplitudes under anodal and cathodal DCS application during ω-agatoxin administration were different compared to the no-DCS condition [D = 0.18, *p* < 0.0001 and D = 0.04, *p* = 0.0005 in K-S for both anodal and no-DCS and between cathodal and no-DCS]. N = 6 mice, n = 22 neurons for all panels. A maximum of two neurons were recorded per slice. ns = non-significant.

**Figure 3 ijms-24-16866-f003:**
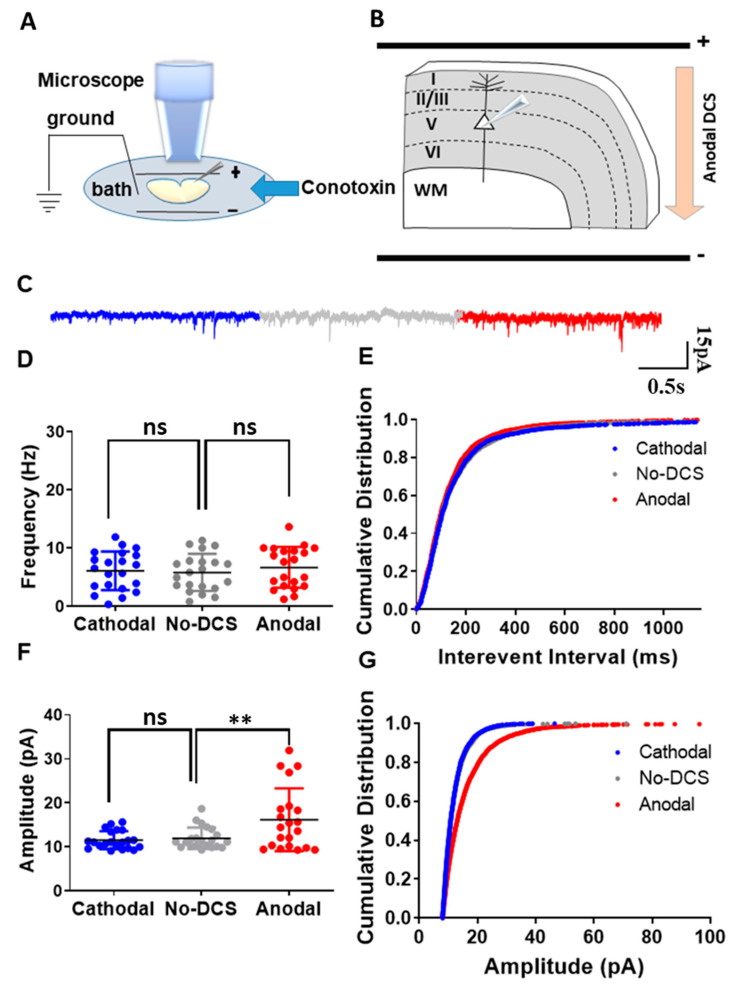
N-type calcium channels are involved in the DCS-induced modulation of presynaptic vesicle release. (**A**) Illustration of the experimental setup. Pyramidal neurons from layer-V M1 motor cortex were recorded when DCS (5 V/m) was applied across the slice and ω-conotoxin was perfused via bath. (**B**) The diagram depicts the direction of the electrical field with respect to the orientation of the recorded layer-V cortical pyramidal neurons, via an example of anodal DCS application. (**C**) Sample traces for sEPSCs recorded under anodal, cathodal, and no DCS conditions are displayed in red, blue, and grey colors, respectively. (**D**) The application of ω-conotoxin did not induce any significant alterations in the mean sEPSC frequency during anodal and cathodal DCS in comparison to no-DCS sEPSC frequency [F_(1.40, 28.06)_ = 1.30, *p* = 0.18 in RM-ANOVA; t_(20)_ = 1.35, *p* = 0.58 and t_(20)_ = 0.85, *p* > 0.99 in posthoc Bonferroni corrected comparisons between anodal to no-DCS and between cathodal to no-DCS, respectively]. Data are presented as Mean ± SD. (**E**) However, DCS application during administration of ω-conotoxin asymmetrically altered the cumulative distributions of the sEPSC interevent intervals. During anodal DCS application, the cumulative distribution of interevent intervals was observed to shift leftward despite the ω-conotoxin administration, while the cumulative distribution of the interevent intervals remained comparable to the no-DCS condition during cathodal DCS [D = 0.09, *p* = 0.001 and D = 0.06, *p* = 0.08 in K-S for both anodal and no-DCS and between cathodal and no-DCS]. (**F**) Anodal-DCS enhanced the mean sEPSCs amplitude despite ω-conotoxin administration, while cathodal-DCS did not alter the mean sEPSCs amplitude compared to no-DCS condition [F_(1.09, 21.80)_ = 11.37, *p* = 0.002 in RM-ANOVA; t_(20)_ = 3.21, *p* = 0.01 and t_(20)_ = 0.85, *p* = 0.55 in posthoc Bonferroni corrected comparisons between anodal to no-DCS and between cathodal to no-DCS, respectively]. Data are presented as Mean ± SD. (**G**) Under ω-conotoxin application, the cumulative distribution of the sEPSCs amplitudes was observed to shift during anodal DCS application, while the cumulative distribution of the sEPSCs amplitudes was comparable to the no-DCS condition during cathodal DCS application [D = 0.22, *p* < 0.0001 and D = 0.02, *p* = 0.61 in K-S for both anodal and no-DCS and between cathodal and no-DCS]. N = 5 mice, n = 21 neurons for all panels. A maximum of two neurons were recorded per slice. ** *p* < 0.01; ns = non-significant.

## Data Availability

The data presented in this study are available on request from the corresponding author.

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
