# Peer review of "Direct Current Stimulation Modulates Synaptic Facilitation via Distinct Presynaptic Calcium Channels"

_ijms, 2023, doi:10.3390/ijms242316866_

Round 1

Reviewer 1 Report

Comments and Suggestions for Authors

In this manuscript, the authors explore how DCS stimulation modulates presynaptic calcium channel conductance through the monitoring of synaptic vesicle release. They specifically examine P/Q, N, and T type calcium channels using specific inhibitors and propose the involvement of P/Q and N type calcium channels in DCS stimulation from layer V pyramidal neurons.

Comments:

1. In the EPSC recording section, have you added a sodium channel blocker? Cation flux signals can affect recordings, and including this step is crucial to ensuring data accuracy.

2.Regarding the statistical analysis, in Line 166, you mentioned F(1.3 37.8). Please clarify the meaning of this range and explain what t(29) represents in the statistical analysis. Provide more detail on the statistical methods employed.

3. In Line 171, you mention D=0.09. Please specify what "D" represents and provide a brief explanation for its relevance in the context of your study.

4. Explain the physiological significance of the amplitude differences. Typically, activity is represented using frequency, but it's important to discuss why amplitude differences matter in this context.

5. I noticed that Figures 1 to 3 all depict conditions with inhibitor perfusion. Do you have data for the same experiments conducted without inhibitors? Including this comparison could provide valuable insights.

6. In Lines 294, 299, 301, and 305, there are references to "@-conotoxin." Please correct the formatting or provide the correct symbol and notation.

7. In Lines 227 and 230, you mentioned "@-agatoxin." Please ensure correct formatting or provide the accurate symbol and notation.

8. In Line 503, you mention "300@m thickness." Please correct the formatting for clarity.

9. Is there a specific reason for using male mice exclusively in your experiments? Address the rationale for this choice and any potential implications for the study's outcomes.

10. Have you investigated the expression of L-type calcium channels in pyramidal neurons? If relevant to your study, please provide a brief discussion or mention any findings related to L-type channels. or please list literature related with L type expression. 

Reviewer 2 Report

Comments and Suggestions for Authors

This paper represents a continuation of the authors’ previous studies investigating the effect of different voltage-gated ion channel blockers on tDCS, on this occasion in cortical neurons. This comes with a mixture of strengths and weaknesses. The present paper is concise and reports effects of three different voltage-gated calcium channel (VGCC) blockers on tDCS in cortical neurons, thus extending work to ion channel subtypes. However, there are weakness in the range of channel blockers investigated and, in particular, a lack of illustrated control recordings, as detailed further below.

Major points

1.     It is unclear how layer V cortical neurons are recorded from hippocampal coronal brain slices? Are these Methods copy and pasted from a previous publication?

2.     A major omission is the lack of control recordings. This is partly mitigated by reports that the T-type channel blocker used may have “no effect” on DCS-induced changes (but see below), however, the authors cannot state that the drugs used “occlude” effects when no control phenotype is illustrated.

3.     Related to point 2, are these effects reversible?

4.     The authors state throughout that the T-type channel block used has “no effect”; however, this is not correct. ML-218 has reported actions on cathodal tDSC. These actions are in fact equivalent to this measure for agatoxin and conotoxin (as above, control recordings are needed in all cases). In some cases, the authors state that ML-218 has no effect on sIPSC frequency modulation. In this regard, it is common to equate modulation of sIPSC frequency with presynaptic vesicle release, whilst sIPSC amplitude may be considered a post-synaptic phenomenon. The paper would be improved by a more nuanced discussion of what effects on each parameters likely mean.

5.     Related to this point, given the lack/equivocal effect of ML-813, it is recommended that an additional T-type blocker (such as the more common TTA-P2) be used to confirm that these effects are not drug specific.

6.     As above, this is concise study. A clear question, alluded to in the Discussion, is the role of R-type (and also L-type) VGCCs. The study would be improved by including additional blockers to get a full picture of subtype effects.

7.     Perhaps related to the points above, the Discussion feels very repetitive and could be broadened. For example, the study suggests functional redundancy between P/Q and N-type VGCCs, but these aspects are not considered. 

8.     Related to point 7, there is good evidence that developmental switches in presynaptic VGCC subtype expression/function can occur throughout the CNS. Here, mice are stated to be “2 months” old – firstly, this is very imprecise and unusual (all mice were exactly 2 month old?) a more accurate range of days should be given; secondly, do the authors consider this age to present an adult phenotype?

9.     Does tDCS have effects on mEPSPs in this system?

10.  A more precise description of replicates should be given. It is recommended to state the number of animals and number of brain slices- if is unclear at present if multiple slices came from one animal.

11.  Figures would benefit from showing sample traces in all figures.

Minor points

Several words have become hyphenated in the submitted version.

Definitions occasionally deviate (tDCS vs DSC)

Reviewer 3 Report

Comments and Suggestions for Authors

The article by Vasu et al. describing "Direct current stimulation modulates synaptic facilitation via distinct presynaptic calcium channels" is a well executed study to determine importance of transcranial direct current stimulation in cortical V layer upon treatment with calcium channel inhibitors to decipher the type of calcium channel and the underlying mechanism regulating the release of pre-synaptic vesicles upon stimulation. The study design is logical and authors present the results in convincing and apt manner. 

However, authors do not discuss about the importance of these findings in light of tripartite synaptic structure. Many studies have established the role of astrocytes in the synapse formation and information transfer between synapses. Have authors looked at the impact of calcium channel blockage on gliotransmitter release by astrocytes and overall impact of occlusion of DCS effect. If not are the authors planning to perform that experiment?

Minor issue

There are certain places in the manuscript where greek alphabets are not properly printed. e.g. page 9 lines 291, 294 and multiple lines on page 10.

Round 2

Reviewer 1 Report

Comments and Suggestions for Authors

The author has addressed all of my questions, and I have no further inquiries regarding this manuscript. Thus far, it appears to be in good shape.

Author Response

Thankful for the time and attention he could afford towards reviewing our manuscript